# Detection of the 1016Gly and 989Pro Knockdown Resistance Mutations in Florida, USA *Aedes aegypti*

**DOI:** 10.3390/insects15110863

**Published:** 2024-11-04

**Authors:** Alden S. Estep, Neil D. Sanscrainte, Frieda Lamberg, Darrel McStoots, Susan Gosselin

**Affiliations:** 1Mosquito & Fly Research Unit, Center for Medical, Agricultural and Veterinary Entomology, United States Department of Agriculture, 1700 SW 23rd Drive, Gainesville, FL 32608, USA; neil.sanscrainte@usda.gov; 2Entomology and Pest Control Section, Bureau of Scientific Evaluation and Technical Assistance, Division of Agricultural Environmental Services, Florida Department of Agriculture and Consumer Services, 3125 Conner Boulevard, Tallahassee, FL 32399, USA; frieda.lamberg@fdacs.gov; 3Public Works Department, Osceola County, 1 Courthouse Square, Suite 3100, Kissimmee, FL 34742, USA; darrel.mcstoots@osceola.org (D.M.); susan.gosselin@osceola.org (S.G.)

**Keywords:** *Aedes aegypti*, knockdown resistance, V1016G, S989P, Florida, USA, *kdr*, insecticide resistance

## Abstract

In *Aedes aegypti*, resistance to pyrethroid insecticides is widespread and has been strongly linked to the presence of mutations in the voltage-gated sodium channel. Distinct ensembles of these mutations are found in the Western Hemisphere and Indopacific with little known mixing. The Indopacific mutations have not previously been detected in the USA. We detected the presence of the V1016G and S989P mutations in three recent collections from Osceola County, Florida. These findings were confirmed using multiple methods and represent a significant expansion of the geographic range of these resistance factors. This requires the modification of the existing resistance screening protocols and further work to understand the operational implications for mosquito control.

## 1. Introduction

*Aedes aegypti* L. is the primary vector of several arboviruses that are not effectively controlled by vaccines, and, therefore, the effective vector control of this species is a critical element of maintaining public health for much of the at-risk population in the tropics and subtropics. Eradication efforts, beginning officially in 1947 by the Pan-American Health Organization, were initially successful in many countries, but these efforts ultimately failed and *Ae. aegypti* reinvaded many regions as well as invaded new areas [1,2]. These control efforts primarily relied on intense chemical control methods, and this, combined with the native plasticity of *Ae. aegypti,* led to rapid development of insecticide resistance [1,3,4].

Early reports of insecticide resistance (IR) and the desire to maintain effective control have led to much study to understand the mechanisms responsible for the resistant phenotype [5,6,7]. While numerous mechanisms may contribute to IR in an organism, the importance of individual mechanistic contributions is dependent on the specific active ingredient being considered [8]. Both enzymatic resistance and target site resistance have been shown to contribute to IR in *Ae. aegypti*, and a strong linkage genetic basis for resistance to dichlorodiphenyltrichloroethane (DDT) and organophosphates was indicated [5,9]. Resistance developed quickly after the widespread introduction of pyrethroids, which, like DDT, also target the sodium channel (NaV). This target site resistance is known as knockdown resistance (*kdr*) and is the result of inherited SNPs in the coding sequence of the NaV that are selected under pressure [10,11,12]. *Kdr* has been demonstrated in numerous insects and, while specific *kdr* SNPs vary between insects and even mosquitoes, *kdr* has been shown to result in the distinct modulation of the native NaV and has been shown to reduce the efficacy of specific classes of pesticides [13,14].

Numerous *kdr* mutations have been described in *Ae. aegypti*, but the functional effect of many of these mutations is unclear. However, two *kdr* ensembles that have been demonstrated to result in intense pyrethroid IR in field populations are the combination of SNPs resulting in an isoleucine at position 1016 (1016I) and cysteine at position 1534 (1534C) and the combination of SNPs resulting in a proline at position 989 (989P) and glycine at position 1016 (1016G) [15,16,17,18]. Interestingly, these two ensembles of SNPs have maintained geographic separation with the 1016I/1534C combination widely found in the Western Hemisphere and the 989P/1016G combination found in the Indopacific [18,19,20]. The 1534C SNP is present in the Indopacific, but studies have demonstrated that it results in moderate levels of IR [21]. The 1534C mutation is usually in opposition to the haplotype containing 989P/1016G in the Indopacific [18,20,22,23].

This geographic stratification of *kdr* mutations is not absolute and a few examples of mixing have been identified. The first detection of the 989P/1016G in the Western Hemisphere occurred in Panama and may represent a temporary infestation associated with international trade through the Panama Canal as it has not been detected subsequently [24]. In 2023, 1016G was detected in West Africa, although the lack of experimental controls for the likely presence of 1016I makes for inconclusive results [25]. Notably, contemporaneous studies in neighboring countries have identified 1016I in Ghana in 2013 and Burkina Faso in 2019, while 1016G was not detected in Nigeria in 2021 [26,27,28]. Interestingly, 989P has been detected without the presence of 1016G in West Africa [28].

To our knowledge, the 989P and 1016G ensemble has not been found in the United States or Mexico, although it is seldom considered in testing because it is expected to be absent based on the stratification of the *kdr* ensembles. However, a study from New Mexico did examine S989P and did not identify the presence of the 989P SNP, which would imply that 1016G was absent (although this was not specifically tested), since the 989P mutation is usually only found when 1016G is found [18,29]. A study in the Florida Keys did not find 1016G, although, like several of the studies cited above, allele-specific controls for the 1016G SNP were not included in the assay [30].

Our study, which examined recent infestations of *Ae. aegypti* in one county along the central spine of Florida, assessed the *kdr* genotype for three SNPs: V1016I, V1016G, and F1534C. When initial melt curve assay testing identified the presence of samples containing 1016G, we confirmed this by AS-PCR, Sanger sequencing, and next-generation sequencing (NGS), which also allowed us to assess S989P. These methods confirmed that the 1016G and the 989P SNPs were present in Florida at these locations and this finding may represent the first detection of Indopacific *kdr* mutations in North America. We also consider the impact that this finding should have on *kdr* screening programs that usually assume a geographic separation between the Western Hemisphere and Indopacific *kdr* ensembles.

## 2. Materials and Methods

### 2.1. Arthropod Surveillance Collection Procedures

Mosquito collections were performed by Osceola County Public Works employees as part of normal vector control activities and collected with the permission of the property owners. Samples were collected using BG Sentinel traps placed at commercial automobile tire shops (GPS locations: 28.245, −81.226; 28.258, −81.457; and 28.305, −81.349). Collected mosquitoes were allowed to desiccate, identified in terms of species, and then aspirated into 2 mL microcentrifuge tubes before shipping at ambient temperature to USDA-ARS-CMAVE in Gainesville, FL. Samples were collected between 4 March and 19 March 2024.

### 2.2. Control Strains

Haplotype-specific controls for assays were provided by well-described laboratory strains reared under standard conditions in the insectaries of USDA-ARS-CMAVE. The Rockefeller (ROCK) strain is permethrin-susceptible and has no known *kdr* mutations (genotype 989SS/1016VV/1534FF, or SSVVFF in shorthand, based on the standard *M. domestica* sodium channel notation) [31]. The Puerto Rico (PR) strain (989SS/1016II/1534CC, or SSIICC) is pyrethroid-resistant and representative of the Western-Hemisphere *kdr* ensemble generally found in the Americas, Caribbean, and West Africa [32]. The cytochrome-and *kdr*-resistant (CKR) strain (989PP/1016GG/1534FF, or PPGGFF) is representative of the pyrethroid-resistant Indopacific *kdr* ensemble and was kindly provided by Dr. Jeffrey Scott [33,34].

### 2.3. Detection of Knockdown Resistance Mutations by Melt Curve Assay

Standard melt curve assay protocols were used for the detection of the V1016I, V1016G, and F1534C *kdr* SNPs from unpurified homogenates using previously described primers and methods [15,35,36,37]. Briefly, individual mosquitoes were placed in 2 mL deep-well homogenization plates with 400 µL deionized water and 1.0 mm zirconium homogenization beads. A control for each allele under test was included in the assay and consisted of the following: ROCK strain (989SS/1016VV/1534FF, haplotype SVF), Puerto Rico strain (989SS/1016II/1534CC, haplotype SIC), Scott CKR strain (989PP/1016GG/1534FF, haplotype PGF), an artificial heterozygote created by combining a ROCK and PR (haplotypes SVF and SIC), and another heterozygote created from a PR and CKR (haplotypes SIC and PGF). Artificial heterozygotes were included to ensure that MCA conditions were optimized for the identification of heterozygous genotypes. A negative control with no mosquito was also included. Raw MCA data are in the data repository.

Sample plates were sealed and then homogenized for 60 s at 30 hertz (Omni International, Kennesaw, GA, USA), spun for 2 min at 2500 rpm, and then maintained on ice until assay setup. Assays were assembled in 384-well plates on an Eppendorf EpMotion 5750 workstation in a final volume of 10 µL as described previously [15,35]. Assembled reactions were amplified for 45 cycles on QuantStudio6Flex (ThermoFisher, Carlsbad, CA, USA) using SYBRgreen chemistry. Melt curve phase data were collected continuously from 90–95 °C. Presence of a particular SNP was called when the melting temperature (T_m_) of the unknown sample was ±0.3 °C of the T_m_ of the appropriate control for the same allele.

### 2.4. DNA Purification from Sample Homogenates

DNA was purified from 150 µL of select homogenates using a silica spin column kit (ZymoResearch, Irvine, CA USA) and following the manufacturer instruction for liquid samples. The samples purified included the controls described above and the eight samples that gave indication of the presence of 1016G in the initial melt curve assay. DNA was eluted in 50 µL of DNA elution buffer and then used for the confirmatory studies described below.

### 2.5. Detection of Knockdown Resistance Mutations by Allele-Specific PCR

Allele-specific PCR for the 1016 region (M2 primers) was conducted using the primers of Li et al. [38]. Duplicate reactions were assembled in strip tubes. Both reactions included the amplicon forward and reverse primers and then one of the two allele-specific primers. Amplification was conducted (94 °C for 3 min, 40 cycles of 94 °C for 30 s, 60 °C for 30 s, and 72 °C for 1 min) with an annealing temperature of 60 °C. After cycling, 5 µL of each reaction, along with 100 bp ladder, was loaded onto a 1% agarose gel with SYBRsafe and electrophoresed at 93V for approximately 1 h. Gels were visualized on an iBright imaging system and exported as image files. Untrimmed gel images with molecular weight standards are in the data repository.

### 2.6. Detection of 1016 Knockdown Resistance Mutations by Sanger Sequencing

Samples were amplified from purified DNA as above using only the outer forward and reverse M2 primers. Unpurified PCR product was shipped to Psomagen, Inc. (Rockville, MD, USA) for ExoSAP purification and Sanger sequencing. Chromatograms with low background in the region of the 1016 *kdr* mutations were visually examined and called for 1016 *kdr* alleles. Heterozygotes were identified by two in-phase overlapping peaks of similar height and well above background noise levels at the site of the SNP. Sanger sequencing chromatograms are in the data repository.

### 2.7. Detection of Knockdown Resistance Mutations and Speciation by Nanopore Sequencing

Purified mosquito DNA, amplified with M1, M2, M3, and COI primers, as in Section 2.5 (COI annealed at 55C, M1 at 51.4C, and M2 and M3 at 60 °C), was prepared for Nanopore sequencing by mixing 2 µL of the M1, M2, M3, and COI PCR products from each sample [38,39]. Four microliters of this unpurified mixture of PCR products from each sample was used as input into the SQK-NBD114.96 sequencing kit (Oxford Nanopore Technologies, Oxford, UK) and each sample was given a different barcode. Protocol NBA_9170_v114_revH_15SEP2022 update 5 April 2023 was used to prepare and load the flow cell. Sequencing was conducted on a Minion Mk1B attached to a DELL workstation with A5000 GPU running Ubuntu 20.04. MinKNOW (Version 23.11.7) software managed basecalling and sequencing operations. Sequencing time was 72 h and a minimum quality filter of 10 was used for passing reads to avoid low-quality base calls.

### 2.8. Bioinformatic Analysis of Nanopore Sequencing Data

After sequencing, the sequencing summary file was filtered to ensure only the highest quality sequences were used for subsequent analysis. Filtering required the presence of a barcode region (>37 bases in length and >95% identity) at each end. Sequences meeting filtering minima were separated by barcode and selected by seqtk [40]. These high-quality sequences for each sample were mapped using Minimap2 [41,42] using a model index containing the *Ae. aegypti* COI (KY022526.1) and the 989, 1016, and 1534 regions of the sodium channel (NC_035109.1:315938900-315939599 and NC_035109.1:315983000-315984499). Genotype calls were made by examining the mapped reads in IGV [43]. Coverage of less than 100×× for any amplicon resulted in the exclusion of the sample from further analysis.

## 3. Results

### 3.1. Effect of Non-Target 1016 Alleles in the V1016I and V1016G Melt Curve Assays

The standard MCAs in wide use for the 1016 *kdr* mutation differentiate between two alleles, 1016V and 1016I or 1016V and 1016G [36]. We could find no study that examined the effect of a 1016G allele present in the V1016I assay or a 1016I allele present in the V1016G assay so we included controls for all three possible alleles in each assay. Both assays showed that the non-target allele did produce a result that could obscure the true genotype. In the V1016I MCA, the 1016G allele resulted in a melt curve and T_m_ that matched the 1016V allele (Figure 1A). Notably, there was only a slightly increased T_m_ in the 1016I versus 1016V but it was within the standard ± 0.3 °C range around the expected T_m_ so 1016G would be indistinguishable from 1016V. In the V1016G MCA, the 1016I allele resulted in a T_m_ that was 0.8 °C lower than that of the 1016V allele (Figure 1B).

Based on these findings, the V1016I MCA produces two outcomes; it unambiguously establishes the presence of the 1016I allele and the presence of a 1016V or a 1016G. Similarly, the V1016G MCA produces two outcomes; it establishes the unambiguous presence of the 1016G allele and the presence of a 1016V or a 1016I. To unambiguously determine the two alleles present at 1016 in a particular sample, both V1016I and V1016G assays would have to be conducted.

### 3.2. Assessment of V1016I, V1016G, and F1534C in Osceola County, FL by Melt Curve Assay

Melt curve assay methods were used to assess the presence of the V1016I, V1016G, and F1534C in three collections of *Aedes aegypti* in Osceola County, Florida. Homozygous allele controls, and negative controls gave expected results, indicating a valid assay (Figure 1A,B). One heterozygous control (VIFC) gave the expected result (Figure 1A) but the other (GIFC) did not indicate the presence of the 1016G allele (Figure 1B). As expected from previous studies in US *Ae. aegypti*, the 1534C allele was common across all three populations (Figure 2) [15,29,30,35,44,45,46,47]. Only 9 of the 111 samples tested were heterozygous for the 1534F allele. The 1016I allele was also common; 109 of the 111 samples had at least one copy and 73% were homozygous. The V1016G MCA indicated the presence of the 1016G allele in samples (8 of 111) from the three locations. The melt curve assay results determined that seven samples were the GIFC genotype and were found in all three locations. One VGFC sample was detected in the Osceola 3 collection.

### 3.3. Confirmation of the Presence of 1016G in Florida Aedes aegypti by Allele-Specific PCR

To confirm the detection of 1016G, we purified DNA from a subset of Osceola County sample homogenates. This included all the samples with an indication of 1016G as well as the controls from the MCA. Purified DNA was used with the M2 AS-PCR primers for the 1016V and 1016G *kdr* alleles [25,38]. As with the 1016 MCAs, we could find no study that evaluated the result of the presence of the 1016I allele in this AS-PCR system. Our IICC control indicated that the presence of the 1016I allele can result in weak amplification in the 1016V AS-PCR assay (Figure 3, top, lane 9 with weak 1016V shown as grey letter). Weak 1016V bands were also observed in lanes 2, 4, and 5 and these likely represent this same ambiguity from the presence of 1016I as indicated by MCA. While only 6 of the 10 Osceola samples gave a result in this AS-PCR, 5 of those confirmed the identification of 1016G (Figure 3, bottom).

### 3.4. Confirmation of the Presence of 1016G by Sanger Sequencing

The M2 amplicon PCR product from each of the 18 samples was shipped to a commercial Sanger sequencing provider to determine the alleles present at the 1016 *kdr* location (Table 1). Sanger sequencing confirmed the presence of 1016G in four of the eight Osceola samples identified by MCA; the other four 1016G MCA positive samples did not produce a result. All homozygous allele controls produced the same result as the 1016 MCA assays. Notably, the expected GIFC control that MCA identified as IIFC was also II by Sanger sequencing, indicating that the MCA call was correct.

### 3.5. Assessment of 989, 1016, and 1534 kdr Genotypes and Speciation by Next-Generation Sequencing

We conducted Nanopore amplicon sequencing on 16 of the 18 samples for four reasons: to confirm the presence of 1016G in these Florida samples, to confirm the MCA-based 1016 and 1534 genotyping, to expand our investigation to assess the presence of the 989P *kdr* SNP (which is often found with 1016G and results in a strongly pyrethroid resistant genotype [18]), and to ensure, by molecular barcoding, that all samples tested were *Ae. aegypti*. We amplified the purified DNA from each sample with the M1, M2, and M3 *kdr* region primers as well as the COI sequence with standard Folmer primers. After combining the four amplicons for each sample, performing the sequencing and quality control filtering, reads were mapped to the appropriate genomic regions and assessed for the presence of *kdr* mutations (Figure 4). Four samples (including the nuclease-free water negative control sample) did not result in adequate read coverage and were excluded from analysis. Controls gave the same 1016 results as initially determined by MCA. Six of the eight Osceola 1016G-containing samples identified by MCA were confirmed by NGS. Two 1016G-containing samples did not produce sufficient coverage so no call was made.

While the coverage of the 989 region was lower than the other *kdr* regions, it was more than sufficient to confidently identify the presence of the 989P SNP. The CKR-derived GGFF control was also homozygous for 989P, matching the previously published CKR genotype, while the VVFF and IICC controls were 989S, as expected. Among the field collections, we observed 989P in three of the six 1016G-positive Osceola samples (Table 1).

Cytochrome oxidase I consensus sequencies were extracted from the alignment files and assessed for identity by BLASTn [48]. All 14 successfully sequenced samples were identified as *Ae. aegypti* with 12 of 14 matching at greater than 99% (Table 1). Of the two that had a less than 99% match, one matched at 98% and one at 98.9%.

## 4. Discussion

### 4.1. Identification of 1016G and 989P in the Continental US

The primary finding of this study is the identification and confirmation of the 1016G and 989P *kdr* mutations in recent collections of *Ae. aegypti* from three locations within Osceola County, FL. To our knowledge, this represents the first detection of these mutations in North America. Notably, the 1016G SNP was infrequent and only detected as a heterozygote by MCA; in seven of the eight organisms, it was paired with the 1016I allele and was present at about 10% of the Osceola 1 population and less than 10% in the other two Osceola populations. Due to the relatively small number of organisms collected, we were not able to assess if the frequencies differed between populations. This finding immediately raises three questions.

First, we wonder if this detection provides more evidence to support a breakdown in the geographic separation between the Western-Hemisphere *kdr* ensemble linked to strong pyrethroid resistance (the homozygous 1016I and 1534C) and the Indopacific *kdr* ensemble of strong resistance (the homozygous 1016G and 989P) [49]. Recent literature does indicate that there may be some expansion in the range of the 1016G with detections in Saudi Arabia published in 2017 [50]. Subsequently, 1016G was identified in Panama in 2019, West Africa in 2023, and, now, in this study, here in the continental US [24,25].

Second, we wonder if these recent findings of 1016G are not indicative of expansion but instead an artifact of *kdr* surveillance studies that do not expect 1016G to be present. It is possible it has been present at a low level but just not detected. We show, in this study, that both MCA and AS-PCR can give an erroneous result when an unexpected allele is present. As the V1016I MCA is the most common method used to assess the presence of 1016 SNPs in studies in the Americas and Caribbean, it is quite possible 1016G has been detected, but, with the small (~0.3 °C) Tm difference between a true 1016V and a 1016G, it could easily be missed and called as 1016V.

Third, we wonder if our detection of 1016G is temporary, much like travel cases of dengue which cause limited foci of local transmission and infection that do not become endemic. Similarly, all three locations in Osceola where 1016G was found are recent *Ae. aegypti* infestations and all three locations are involved in the international used tire trade, a proven mechanism for moving *Aedes* eggs [51,52,53]. The possible temporary introduction of resistance alleles has been observed in Panama, where recent studies have found no further evidence of 1016G even though it was identified in 2019, although it is quite possible that the sampling strategy may have not been intense enough to detect their presence at a very low level [54,55]. It is certainly possible that the *kdr* SNPs we detected in Osceola County, FL were due to a specific importation event that may not lead to establishment, but this will require further surveillance in this area to assess the potential establishment of these non-native *kdr* mutations and surveillance in surrounding areas of Florida to the current distribution of these mutations.

### 4.2. Operational Implications of the Introgression of Indopacific kdr Mutations

While only the continued sampling with the specific assessment for 1016G and 989P will bring clarity to the questions raised above about the possible spread and establishment of Indopacific *kdr* SNPs into new areas, we think it is wise to consider the impacts of this introgression on operational interventions. The Western Hemisphere already has a *kdr* ensemble that results in a strong resistance to pyrethroids. The 1016I and 1534C genotype (IICC) has been shown to result in longer survival in laboratory bioassays, to lead to the failure of permethrin-treated repellent fabrics and the reduced efficacy of formulated pyrethroid products, and has been shown to result in greater resistance intensity as the frequency of the genotype in the population increases [15,17,45,56]. The 1016G and 989P ensemble has also been shown to result in strong pyrethroid resistance in both laboratory and semi-field studies [18,23,33]. Like the IICC genotype, the presence of the 989P and 1016G ensemble reduces the efficacy of the formulated pyrethroids [18]. The impact of the Indopacific *kdr* SNPs on permethrin-repellent materials like military uniform fabrics has not, to our knowledge, been tested, but is important information.

One concern about the cocirculation of two distinct ensembles of SNPs is that it sets the conditions required for a sudden increase in resistance intensity. Resistance theory holds as a principal that independent mutations with some selective benefit will eventually come together to produce combinations with a greater benefit when under selective pressure [57]. This was demonstrated with distinct subgroups of Indopacific *kdr* SNPs that resulted in an over 1000-fold resistance to deltamethrin when they came together [58]. In addition, selective pressure within a population varies at the individual level based on differing genotypes favoring the survival of genotypes with a higher advantage [59]. A beneficial allele (i.e., providing benefits under selection) can spread extremely rapidly if the selective benefit is large. This was observed in the field when the 1016I SNP entered Peruvian populations of *Aedes aegypti* in which the 1534C SNP was already present [16]. These samples from Osceola County have a high level of the IICC genotype, and, now, we find the 1016G and 989P ensemble. Samples from Osceola County that had the 989P, 1016G, 1016I, and 1534C SNPs together in the same organism were found, but, as the samples were dried and non-viable, it was impossible to isolate the actual NaV transcripts to assess if the distinct *kdr* ensembles had recombined or remained on separate transcripts. We were also not able to quantify the phenotypic intensity of the resistance. The risks associated with the recombination of these *kdr* ensembles are unclear but could be disastrous for operational control.

### 4.3. Recommendations for Insecticide Resistance Monitoring Inclusive of the Presence of Indopacific kdr Mutations

A final impact of finding both 1016G and 989P in the US is that it requires resistance researchers and vector control resistance surveillance programs to account for these mutations when conducting IR screening. We show in this study that 1016G will be miscalled as 1016V in the widely used V1016I MCA, that 1016I will be miscalled as 1016V in the V1016G MCA, and that 1016I may be miscalled as 1016V in the M2 AS-PCR. In all three of these situations, a very resistant allele can masquerade as a wildtype allele. In a most extreme example, a population that was 100% homozygous 1016G (for example, the CKR strain used here) would appear to have the same genotype in the V1016I MCA as susceptible strains like the ORL or ROCK strains (100% 1016V) and give a false sense of operational susceptibility.

We, therefore, suggest a few recommendations for existing *Ae. aegypti kdr* testing workflows to accommodate the possibility that Indopacific mutations may be present in the Western Hemisphere. First, the inclusion of controls for each possible allele being tested is critical for validating MCA, AS-PCR, and fluorescence genotyping studies. Reporting the control results for all the possibilities of the SNPs being assessed ensures confidence in the result. For the 1016 *kdr* mutations, this means including a control for the 1016V, 1016I, and 1016G alleles. For 989, 1534, or other SNPs, it means a control for each allele. In our usage here, the ROCK, PR, and CKR strains provide all the possible alleles of the SNPs we were examining. We also suggest the inclusion of heterozygous controls to ensure that competitive primers in MCA are properly balanced. These allele-positive controls should preferably be of the same type as the unknowns, whether homogenized whole organisms from strains with defined genotypes, homogenized legs, or purified DNA. Even synthetic oligos based on genomic sequences can be used if it is not possible to source an organism with the SNP of interest.

Second, *kdr* testing in the Americas should now include an assessment for all three 1016 alleles. If the testing method of choice is MCA, putting the results of the V1016I and V1016G MCAs together for each sample allows the unambiguous determination of the 1016 genotype. For other assay systems, the proper modification requires a demonstration that other alleles do not result in signal.

Third, while insecticide resistance genotyping using surrogate assays is much cheaper than methods like Sanger sequencing or NGS, there are still costs that increase as more alleles need to be assessed. While we recommend the screening for 1016G to now be included, we recognize this will increase both the time and costs of screening. Using the well-known linkages between SNPs can reduce the number of alleles that need to be assessed. In *M. domestica* (traditionally screened for *kdr* by Sanger), one *kdr* SNP is used as a gatekeeper [60]. If a mutation is detected at this SNP, further testing for additional SNPs is indicated. Similarly, in *Ae. aegypti,* both the 1016I and 1016G alleles have strong unidirectional linkage with 1534C and 1534F, respectively. This linkage means that a particular allele present at 1016 strongly implies the allele found at 1534. Further, the 1534C allele is found without 1016I and provides only a modest increase in resistance, thus assessing that 1534 is not a high-value biomarker for insecticide resistance. If the goal of the IR genotyping is to guide operational interventions, it is probably more important to generate information about 1016 than 1534. Samples positive for 1016G can then be screened for 989P.

Alternatively, if maintaining continuity between previous *kdr* screening efforts and current screening efforts is important as it is for our laboratory, IR screening can continue with the standard V1016I and F1534C assays, but, since the 1534F allele is relatively rare and nearly always found when 1016G is found, any sample containing the 1534F allele can undergo an additional screening for V1016G and S989P. In many locations in the Americas, 1534C is fixed or nearly fixed; thus, the additional screening of samples with evidence of 1534F for V1016G and S989P would not add much additional effort to IR screening workflows.

## 5. Conclusions

The identification of the 1016G and 989P *kdr* mutations in *Ae. aegypti* represents a novel finding in the US that has important implications for operational mosquito control. While excellent baseline work has been carried out to quantify the effect of these *kdr* SNPs on phenotypic resistance, much work about the effect on operational interventions is unclear. Also unclear is the phenotypic result when these recently detected SNPs interact with existing *kdr* mutations. What is clear is that the presence of these *kdr* mutations increases the likelihood of a rapid increase in IR so existing workflows need to be modified to account for the presence of these *kdr* alleles.

## Figures and Tables

**Figure 1 insects-15-00863-f001:**
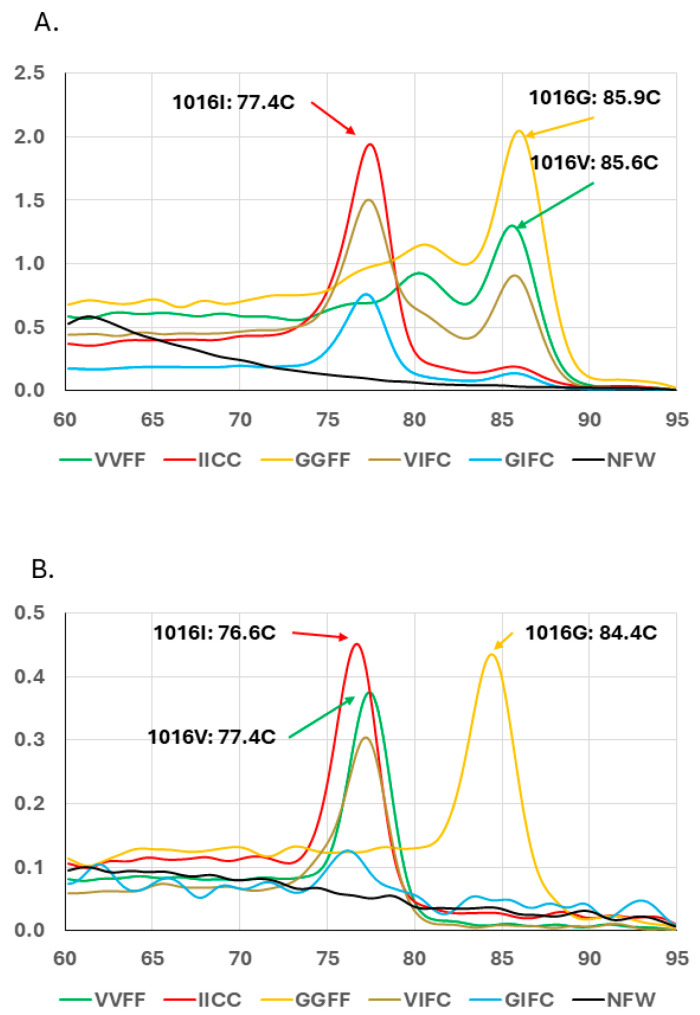
Representative melt curve spectra for the three *Aedes aegypti kdr* 1016 alleles using control samples shows the ambiguity introduced by the presence of the non-targeted allele. (**A**) The 1016G allele appears as 1016V in the standard V1016I melt curve assay. (**B**) The 1016I allele appears as 1016V in the standard V1016G melt curve assay. The horizontal axis is temperature in degrees Celsius and the vertical axis is the derivative of fluorescence. The melting temperature of the 1016V, 1016I, and 1016G alleles are noted for each assay.

**Figure 2 insects-15-00863-f002:**
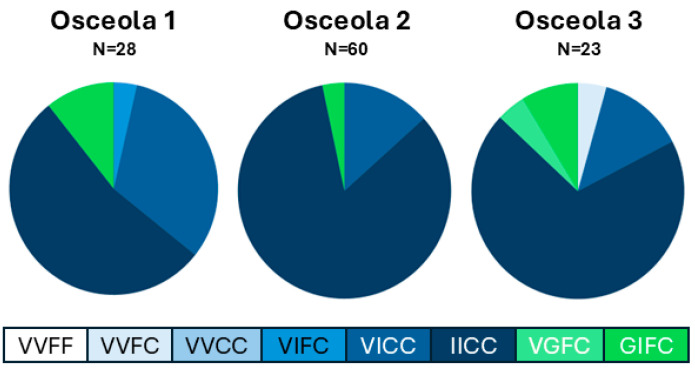
Genotyping of the 1016 and 1534 *kdr* mutations by melt curve analysis indicates the presence of the 1016G haplotype in 3 surveillance collections from Osceola County, Florida. Genotyping was performed by melt curve analysis using previously described primers and methods [15,35,36]. Assay controls included a control for each haplotype, Rockefeller strain (989S, 1016V, and 1534F) [31], Puerto Rico strain (989S, 1016I, and 1534C) [32], CKR strain (989P, 1016G, and 1534F) [33,34], artificial heterozygotes created by homogenizing CKR and PR (989SP, 1016GI, and 1534FC), and ROCK and PR (989S, 1016VI, and 1534FC), and a nuclease free water blank.

**Figure 3 insects-15-00863-f003:**
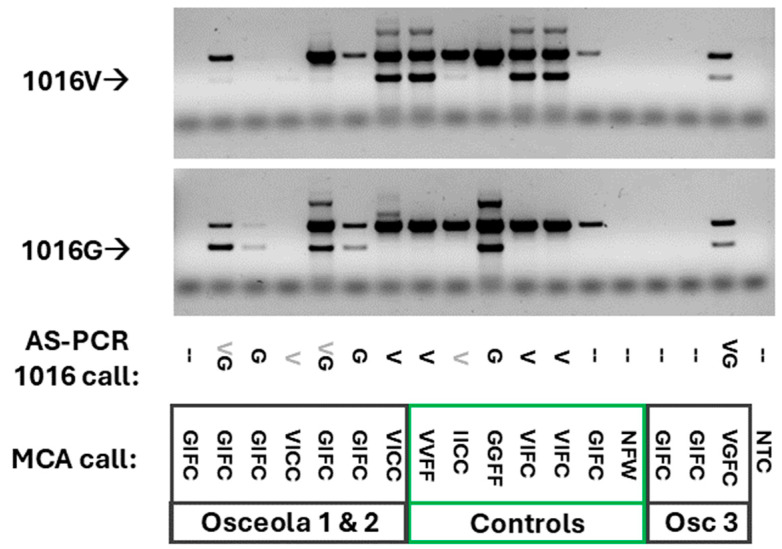
Confirmation of the presence of the 1016G allele in Osceola samples by allele-specific PCR. Purified DNA from controls from samples that indicated the presence of the 1016G allele by melt curve assay was amplified using the M2 primer set for the 1016 *kdr* mutation region [38]. Letters in grey represent a weak detection. Five microliters of each reaction was electrophoresed on a 1% agarose gel and visualized with SYBRsafe. Images were collected, then inverted and increased in contrast to see faint bands. Images were trimmed for presentation. Untrimmed images including molecular weight markers are available in the data repository. Five of the eight Osceola samples were confirmed for the presence of the 1016G haplotype.

**Figure 4 insects-15-00863-f004:**
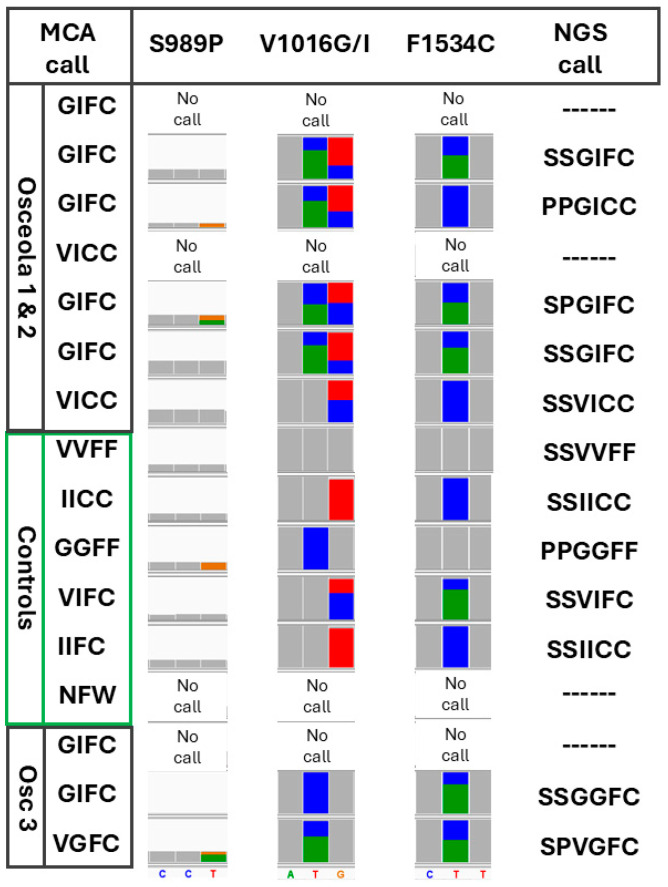
Genotyping by next-generation sequencing demonstrates the presence of 1016G and 989P *kdr* haplotypes in Osceola County, Florida. Purified DNA from controls and samples that indicated the presence of the 1016G haplotype by melt curve assay were amplified using the M1, M2, and M3 primer sets for the 1016 *kdr* mutation regions [38] and the Folmer COI primers [39] for species verification. Sequencing was conducted on a Minion Mk1B using R10 technology. Sequencing run metadata are in the data repository. Sequencing reads, aligned to portions of NC_035109.1, are available from NCBI SRA: PRJNA1170904. Codons for the 989, 1016, and 1534 mutations are shown in the antisense direction. Expected nucleotides are in grey while mutated nucleotides are shown colored (A-green, T-red, C-blue and G-brown). Six of the eight Osceola samples were confirmed for the presence of the 1016G haplotype. Three of the six with 1016G also had 989P. Genotype calls were not made when coverage was less than 100× for any *kdr* SNP.

**Table 1 insects-15-00863-t001:** Genotyping comparison using multiple methods and verification of species identity by cytochrome oxidase I sequencing.

Sample	MCA 1016/1534	AS-PCR 1016	Sanger 1016	NGS 989/1016/1534	COI Accession-%ID
Osceola 1	IICC	no call	no call	not tested	not tested
Osceola 1	GIFC	no call	no call	no call	MN299016.1-100%
Osceola 1	GIFC	weakV G	GI	SSGIFC	MK300224.1-99.72%
Osceola 1	GIFC	weakG	no call	PPGICC	MN299016.1-99.57%
Osceola 1	VICC	weakV	no call	no call	MN299016.1-100%
Osceola 2	GIFC	weakV G	GI	SPGIFC	MK300218.1-98.87%
Osceola 2	GIFC	G	GI	SSGIFC	PP902511.1-99.57%
Osceola 2	VICC	V	no call	SSVICC	MN299016.1-100%
Control-SSVVFF	VVFF	V	VV	SSVVFF	MN299016.1-99.57%
Control-SSIICC	IICC	weakV	II	SSIICC	MN299016.1-100%
Control-PPGGFF	GGFF	G	GG	PPGGFF	MK300224.1-99.72%
Control-SSVIFC	VIFC	V	VI	SSVIFC	MK300224.1-98.03%
Control-SSVIFC	VIFC	V	VI	not tested	not tested
Control-SPGIFC	IIFC	no call	II	SSIICC	MK300224.1-99.72%
Control-NFW	no call	no call	no call	no call	no call
Osceola 3	GIFC	no call	no call	no call	no call
Osceola 3	GIFC	no call	no call	SSGGFC	PP902511.1-99.72%
Osceola 3	VGFC	V G	G	SPVGFC	MN299016.1-100%

## Data Availability

The data supporting the work described in this publication are present in the publication itself, available at https://doi.org/10.15482/USDA.ADC/27183570 or from NCBI under accession: PRJNA1170904.

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
