# Peer review of "Detection of the 1016Gly and 989Pro Knockdown Resistance Mutations in Florida, USA Aedes aegypti"

_insects, 2024, doi:10.3390/insects15110863_

Round 1
Reviewer 1 Report
Comments and Suggestions for Authors
The authors of the paper entitled, “Detection of the 1016Gly and 989Pro knockdown resistance mutations in Florida, USA Aedes aegypti ” provide an interesting overview on knockdown resistance (kdr). The article reports the first detection of the 1016G and 989P knockdown resistance (kdr) mutations in Aedes aegypti using various molecular techniques. The study aims to identify and characterize kdr mutations in Ae. aegypti mosquitoes collected from Osceola County, Florida. The researchers employed melt curve assays, allele-specific PCR, Sanger sequencing, and Nanopore sequencing to detect kdr mutations.
Key Findings:
- Detection of Indo- pacific 989Pand1016G kdr mutations in Ae. Aegypt from the continental US
- Co-occurrence of 1016G and 989P mutations with existing kdr mutations (1016I and 1534C).
I have comments related to some parts of the manuscript but mostly are minor problems in the manuscript that will be important to be clarified by the authors before any final decision.
Author is advised to provide a graphical abstract of kdr SNP.
Restriction enzyme-based methods to identify kdr mutations: In order to detect kdr mutations, the authors employed melt curve assays, allele-specific PCR, Sanger sequencing, and Nanopore sequencing. Restriction enzyme-based methods can be used to identify kdr mutations, though, it requires prior knowledge of the mutation site
Phenotypic correlation:
However, the geographical stratification of co-occurrence of mutation VI/FC in western hemisphere VG/SP in the Indo-Pacific is not absolute. The study confirms the prevalence of 1016G and 989P mutations which is Indo-Pacific in the western hemisphere; does this ensemble of SNP also confer strong IR. The functional impact of these mutations has not been investigated.
Suggestion: Conduct functional studies to determine the level of resistance conferred by the 1016G/989P ensemble in these field populations. This could help to predict the effectiveness of insecticides and guide mosquito control programs in these areas
Has Florida contributed to international trade routes? The international trade may play a role in introducing non-native mutations to new regions. Has the author explored this aspect in depth?
The study does not discuss whether the mutations in Florida are part of a long-term established population or the result of a recent introduction. Longitudinal studies to track the persistence and spread of these mutations over time could clarify whether this is a transient or established phenomenon, providing insight into the dynamics of mutation spread.
84-85: However, a study from New Mexico did examine S989P and did not identify the presence of the 989P SNP which would imply that 1016G was absent.
Author is advised to provide more context about the Indopacific kdr ensemble, such as, In Indo -pacific 989P SNP alone is not present and it is found with 1016G. However, 1016G SNP occur alone even in absence of 989P
Methodological limitations:
Melt curve assays may not be 100% accurate (as described by author)
Author can also add some points on other Methodological limitations:
Sanger sequencing may not detect heterozygous individuals accurately.
Nanopore sequencing has limitations in terms of read accuracy and length
Reviewer 2 Report
Comments and Suggestions for Authors
This is a review of the paper entitled “Detection of the 1016Gly and 989Pro knockdown resistance mutations in Florida, USA Aedes aegypti.” It is important to track the movement of resistance mutations into new areas. It becomes more problematic when detection methods cannot distinguish between two (or more) mutations. Documenting the effect is the first step in gaining a better understanding of the problem and improving our ability to mitigate resistance issues in pest management. In this regards this paper will be an important contribution to the literature.
The sample size is small, so it is perhaps lucky that the authors detected the problem in field collected specimens. However, this is a minor point because the goal of the paper was not accurate estimation of field abundance.
At a guess 10% of the Osceola population with 1016I allele means 3 out of 28 (Line 303). So, the 95% confidence interval is roughly 4% to 27% frequency. (Using PooledInfRate in R and assuming insects were processed individually) This range covers the frequency in the other locations. However, keep in mind that accurate field estimates was not the goal in this research. In this respect, these are awesome results that can be used to make better plans in a study where field estimation is the goal.
Line 43) Please do not use acronyms without identifying them. Several acronyms are missing a definition.
Line 62) I am missing the point of this sentence. The connection between the thoughts is unclear.
Line 76) Appears to be two citation styles.
Line 104) Please confirm (or deny) that there was one mosquito per collection tube.
Line 168) Typically a manufacturer is provided even for something like a kit.
Figure 1) The x-axis scale is integer with three decimal places that are never used. The y-axis scale displays to three decimal places but only one (A) or two (B) decimal places are used. The x and y axes could be simplified with no loss in content.
Line 267) “a strongly pyrethroid genotype” resistant?
Line 302) The purpose in this study was not accurate estimation of frequencies. Despite that, you should calculate a 95% confidence interval for your estimated frequencies. The PooledInfRate in R is one option. There is both an R and an Excel version of the software available in association with Centers for Disease Control. Essentially what you will find is that the small sample size results in large 95% confidence intervals and you might not be able to tell any difference between the different locations. The text can be adjusted accordingly.
Line 314) This is an excellent guess. It is difficult to find what you do not look for. Also, there is a tendency to take a few samples and if not detected to declare it is absent. However, it takes a large sample to detect a rare event. Originally, I had considered that Figure 2 would be better as a table with confidence intervals for each genotype. However, accurate estimation of genotype frequency was not a goal in this paper, so I dropped the idea. The figure is good enough for presence-absence questions.
Line 320) Why would it not become endemic? Is there a larger life history cost to 1016G, or no selection pressure?
Line 325) Yes, maybe. Was the sample size in the cited studies sufficient? A better question, given the sample size and sampling design in the cited studies what is the positivity rate below which it is likely that the genotype is present but not detected? If you want another example: What is the trapping effort needed for California to declare that the Medfly was eradicated? James Carey fought this battle. The alternative you suggest would mean considerable migration of mosquitoes from locations outside the USA that have populations with the 1016G mutation. This suggests that there is either a sampling problem, or this is a quarantine issue (or both). Given the distance between sample sites, are these independent or would tire movement from one site increase the infection probability at a neighboring site?
Line 333) “a kdr ensembles” singular-plural
Line 404) … for insecticide resistance.
